# Transcriptional profiling of lung macrophages identifies a predictive signature for inflammatory lung disease in preterm infants

Debashis Sahoo [1,2,3], Livia S. Zaramela[1,4], Gilberto E. Hernandez[1], Uyen Mai[1,2], Sahar Taheri[1,2], Dharanidhar Dang[1,2], Ashley N. Stouch[1], Rachel M. Medal[1], Alyssa M. McCoy[1], Judy L. Aschner[5], Timothy S. Blackwell[6], Karsten Zengler [1,4] & Lawrence S. Prince [1 ✉]

Lung macrophages mature after birth, placing newborn infants, particularly those born preterm, within a unique window of susceptibility to disease. We hypothesized that in preterm infants, lung macrophage immaturity contributes to the development of bronchopulmonary dysplasia (BPD), the most common serious complication of prematurity. By measuring changes in lung macrophage gene expression in preterm patients at risk of BPD, we show here that patients eventually developing BPD had higher inflammatory mediator expression even on the first day of life. Surprisingly, the ex vivo response to LPS was similar across all samples. Our analysis did however uncover macrophage signature genes whose expression increased in the first week of life specifically in patients resilient to disease. We propose that these changes describe the dynamics of human lung macrophage differentiation. Our study therefore provides new mechanistic insight into both neonatal lung disease and human developmental immunology.

[1] Department of Pediatrics, Rady Children's Hospital, University of California, San Diego, La Jolla, CA 92093, USA. [2] Department of Computer Science and Engineering, University of California, San Diego, La Jolla, CA 92093, USA. [3] Moores Cancer Center, University of California, San Diego, La Jolla, CA 92037, USA. [4] Department of Bioengineering, University of California, San Diego, La Jolla, CA 92093, USA. [5] Department of Pediatrics, Joseph M Sanzari Children's Hospital, Hackensack University Medical Center, Hackensack Meridian School of Medicine at Seton Hall, Hackensack, NJ 07110, USA. [6] Departments of Medicine, Cancer Biology, and Developmental Cell Biology, Vanderbilt University School of Medicine, Nashville, TN 37232, USA. ✉email: lprinceucsd@gmail.com

The fetal to newborn transition marks a truly unique immunological landmark. At birth, newborns rapidly shift from a state of fetal immune tolerance to mount a protective response against extrauterine microbial pathogens[1–3]. Unfortunately, the transition from tolerance to activation also represents a window of susceptibility against microbial infections and inflammatory disease[4,5]. In attempting to provide protection for the vulnerable newborn, cellular and humoral immunity undergoes maturation as the fetus approaches delivery[6]. However, in infants born preterm, the normal timing and coordination of immune development are disturbed[7]. Preterm infants encounter extrauterine microbes and pathogens with an immature immune system, potentially failing to mount an adequate response to microbial and host substances[8]. Not only could immune immaturity increase the risk of infection and sepsis, but inability to regulate the inflammatory response and promote wound healing may also contribute to disease in preterm infants[9–13].

The most common serious complication of preterm birth, bronchopulmonary dysplasia (BPD) results when the premature, canalicular stage human lung fails to develop normally[14,15]. As a result, infants surviving with BPD have fewer functional alveolar capillary units[15], ineffective gas exchange[16], and abnormal lung mechanics[17,18]. While the cellular and molecular mechanisms leading to BPD remain unknown, BPD risk is associated with infection and inflammation[19,20]. Experimental BPD models using microbial products[21,22], inflammatory mediators[23], or increased inspired oxygen[24] each stimulate an inflammatory response and prevent normal lung morphogenesis. Our laboratory and others have linked lung macrophage activation to inflammation and injury in experimental BPD models[25–27]. However, little is known about how human preterm alveolar macrophages develop or their role in BPD pathogenesis.

Here, we have leveraged technological advances in transcriptomics to assemble a unique dataset from over 200 preterm human lung macrophage samples and investigate the molecular basis of human lung macrophage development and potential mechanisms leading to BPD. Studying serial samples during patients' NICU course provided new insight into human macrophage development and BPD disease progression. In addition, we tested the ex vivo response in each sample to comprehensively measure human innate immune function in this unique collection of developing macrophages. Our data identified specific genes that when abnormally expressed early in life increased the risk of a patient developing BPD. We also detected a unique pattern of gene expression in patients resilient to BPD, outlining a potential beneficial program of normal human lung macrophage development. This robust dataset will be a valuable reference for future studies investigating development of human immunity and neonatal disease mechanisms.

## Results

### Transcriptional profiling of preterm infant lung macrophages.
To understand the molecular mechanisms of human lung macrophage development, function, and role in BPD pathogenesis, we conducted a clinical study using isolated tracheal aspirate macrophages from intubated preterm infants born before 30 week gestation. One hundred twenty-eight patients intubated for respiratory distress syndrome and surfactant administration were consented for the study. Patient characteristics for the study are shown in the Supplementary Table and Supplementary Data 1. Subjects had a median gestational age at birth of 26 weeks and mean birth weight of 871 g. Ninety two percent of the patients survived to discharge, with an overall BPD incidence of 73.7%. Using criteria suggested by the National Institutes of Health to

assign BPD severity[28], 7.9% of the patients developed mild BPD, 24.6% moderate BPD, and 41.2% severe BPD. Tracheal aspirate macrophage samples were obtained on the first day of life and weekly if the patients remained intubated. By using recently developed approaches for studying low input samples, we were able to perform transcriptional profiling of 112 patient samples (with each sample divided into control and LPS-treated conditions). As many patients were extubated early in their clinical course, more samples were obtained on day 1 than any other single time point (Supplementary Fig. 1, Supplementary Data 1). The latest time point of sampling was at 23 weeks of chronological age.

Supplementary Fig. 2a shows the gene expression patterns for all samples obtained at all time points. Unsupervised hierarchical clustering of genes with expression changes across the dataset demonstrated the diversity across individual patients, time of sampling, and ex vivo LPS stimulation. To compare our data to other human macrophage expression datasets, we obtained publicly available microarray data from human peripheral blood monocyte-derived macrophages, THP-1 cells, and adult alveolar macrophages via the Gene Expression Omnibus (Supplementary Fig. 2b–d; GSE134312). Comparing gene expression across our samples to other datasets revealed distinct patterns of expression. Therefore, the differential pattern of macrophage gene expression in preterm infants likely represents a unique transcriptional program inherent to developing lung macrophages and also may contribute to the disease susceptibilities of preterm infants.

### Early gene expression profiles correlate with disease.
Using our robust dataset, we initially asked if lung macrophage gene expression on the first day of life correlated with disease outcome. Figure 1a shows the volcano plot of differential gene expression on day 1 between patients that eventually developed BPD (diagnosed at 36 weeks corrected gestational age) and patients without BPD. Differentially expressed genes are listed in Supplementary Data 2. The inflammatory mediator genes CXCL5, CXCL6, CCL3, and CCL20 were more highly expressed in patients going on to develop BPD (Fig. 1b). Genes associated with innate immune activation (TIFA, NFKB2, TSPO, SOCS3) were also higher on day 1 in BPD patients. These data support the hypothesis that perinatal lung inflammation (either in utero or at the time of birth) plays an important role in early BPD pathogenesis. Genes with lower expression in macrophages from patients that eventually developed BPD (Fig. 1c) included the adhesion molecule versican (VCAN), the transcription factor ARID3A, and the transcription elongation factor TCEB2. Increased expression of the inflammatory chemokines CCL3 and CCL20 in day 1 samples from patients that developed BPD was confirmed by RT-PCR (Fig. 1d).

To connect our differential gene expression analysis with molecular processes, we conducted pathway analysis using the Reactome database (Fig. 1e, f, Supplementary Data 3)[29]. In day 1 samples, inflammatory cytokine signaling pathways were highly represented in patients developing BPD (Fig. 1e), again consistent with the role of early inflammatory exposure in BPD pathogenesis. Importantly, while IL-10 is considered an anti-inflammatory cytokine, the "IL-10 signaling" Reactome pathway category includes 39 inflammatory mediators, many of which possess pro-inflammatory functions. Macrophages from patients that did not develop BPD expressed higher levels of genes involved in multiple metabolic pathways, particularly mitochondrial respiration (Fig. 1f). These results are supported by emerging data connecting inflammatory signaling in macrophages with aerobic glycolysis and suppressed utilization of oxidative phosphorylation[30].

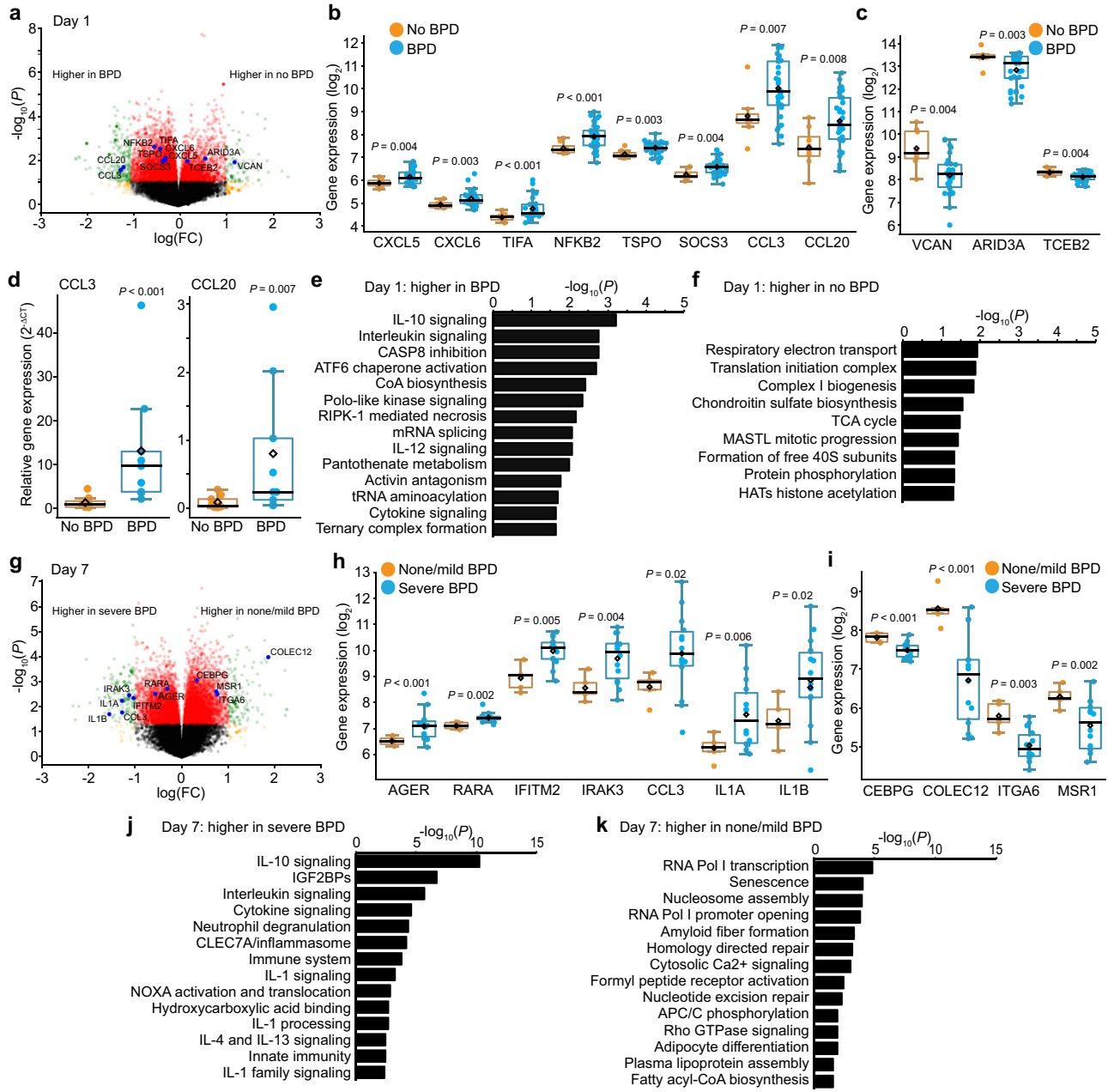

**Fig. 1 Early gene expression profiles correlate with disease. a** Volcano plot showing differential gene expression in tracheal aspirate macrophages isolated on day 1 based on eventual BPD outcome. Genes of interest highlighted. **b** Boxplots with each data point shown, mean as black diamond, median as black line with 95% confidence intervals as colored lines. *P* values calculated by Welch Two Sample *t*-test (unpaired, two-tailed). *CCL3* and *CCL20* were not initially identified using strict FDR, but were statistically different and included based on potential role in BPD pathogenesis. **c** Genes with lower expression in BPD patients. (*n* = 8 for no BPD; *n* = 33 for BPD). **d** Real-time PCR validation of higher *CCL3* and *CCL20* expression in BPD patients. Data shown as mean expression ($2^{-\Delta CT}$). *P* calculated by Mann-Whitney *U* test. **e** and **f** Reactome pathways enriched in samples obtained on day 1 of life. **g** Volcano plot showing differential gene expression in macrophages obtained on day 7 between patients developing no or mild BPD and those developing severe BPD. **h** Selected genes expressed higher in patients eventually developing severe BPD. **i** Selected genes with lower expression in severe BPD patients. (*n* = 5 for none/mild BPD; *n* = 14 for severe BPD). **j** and **k** Reactome pathways enriched in samples obtained on day 7 of life.

We next investigated the developmental dynamics of macrophage gene expression in preterm infants. At day 7 of age, many patients, especially those that did not develop BPD, were extubated and not able to provide a lung macrophage sample. However, we were able to obtain sufficient numbers from patients that either did not develop BPD or had only mild disease (defined as oxygen support for 28 days but no oxygen requirement at 36 weeks corrected gestation). Compared to patients without BPD

or with only mild BPD, preterm infants that eventually developed severe BPD (still requiring positive pressure support or >0.30 $FiO_2$ at 36 weeks corrected gestational age) expressed higher levels of inflammatory genes on day 7, including *IFITM2*, *IRAK3*, *CCL3*, *IL1A*, and *IL1B* (Fig. 1g, h). Severe BPD samples also had higher expression of the receptors for advanced glycosylation end-products (*AGER*) and retinoic acid (*RARA*). Notable genes lower at day 7 in the macrophages of severe BPD patients

included the transcription factor *CEBPG*, cell surface receptors *COLEC12* and *ITGA6*, and the macrophage scavenger receptor *MSR1* (Fig. 1i). Pathway analysis of day 7 samples emphasized the proinflammatory phenotype of macrophages from severe BPD patients (Fig. 1j). Pathways represented on day 7 in patients with no or mild BPD included RNA transcription, cellular senescence, and fatty acid metabolism (Fig. 1k). These analyses further supported the connections between inflammatory macrophage activation and BPD pathogenesis.

**Innate immune function in preterm lung macrophages**. The literature is inconsistent about innate immune capabilities in preterm infants[8]. We directly tested the lung macrophage innate immune response in this study, using the *E. coli* endotoxin and Toll-Like Receptor 4 (TLR4) agonist LPS. Samples obtained on day 1 demonstrated a robust response to LPS (Fig. 2a, b; Supplementary Data 4), expressing high levels of multiple inflammatory genes (*CCL3, CCL4, IRG1, IL6, IL1B, TNF*). Similar responses were measured in day 7 samples, with increased inflammatory gene expression following LPS treatment (Fig. 2c, d). We next tested if the innate immune response to LPS might be different in patients that developed severe BPD compared to those with no or mild disease. Figure 2e shows that LPS similarly induced expression of *IRG1, IL6*, and *IL1B* in macrophages isolated on day 1 from patients with severe BPD and no/mild BPD. LPS also stimulated a robust transcriptional response in day 7 samples (Fig. 2f). While the higher *IL1B* expression in day 7 control samples from severe BPD patients (Fig. 1h, $P = 0.02$) made the LPS-induced fold change lower, the actual gene expression level after LPS treatment was similar between patients with severe BPD and no or mild disease. Collectively these data show that lung macrophages from preterm infants can generate a robust innate immune response to LPS. While patients at increased disease risk expressed higher basal levels of inflammatory genes, the macrophage response to LPS in the first week of life did not correlate with BPD disease severity.

**Expression of macrophage innate immune components**. We next used our dataset to measure how components of the innate immune system might change over time with ongoing human development. We separately examined the expression of select innate immune signaling components (including transcription factors), soluble inflammatory mediators, and cell surface innate immune receptors (Fig. 3, Supplementary Data 5). Data from both control and LPS-treated samples (and LPS:control ratio) were plotted against the chronological week of age. Time points after 10 weeks were not included as sample numbers were lower at later time points due to patients being weaned from mechanical ventilation. A majority of the genes analyzed were expressed at relatively constant levels over time. The key innate immune signaling components *NFKB1, IRAK2, IRAK3*, and *RELA* are indicated in both control and LPS-treated samples (Fig. 3a). In examining expression of soluble inflammatory mediators (mainly cytokines and chemokines), we again observed relatively consistent expression in both control and LPS-treated samples (Fig. 3b). A few select genes (i.e., *TNF, IL1B*) did show an upward trend over time in controls, consistent with higher cytokine expression in patients remaining intubated and requiring mechanical ventilation. The overall patterns of innate immune receptor expression (Fig. 3c) were largely consistent over time, with an exception being the increased expression of the macrophage scavenger receptor *MRC1* after the first day of life. Coupled with the robust LPS response, these data showed that human lung macrophages in preterm infants are immune competent and

express stable levels of innate immune machinery over the first several weeks of life.

**Transcriptional markers of disease resilience**. Because macrophage scavenger receptor expression may correlate with anti-inflammatory and wound healing phenotypes, we examined *MRC1* expression in more detail (Fig. 4a). *MRC1* expression was lowest in day 1 samples. However, in patients resilient to lung disease (no or mild BPD), *MRC1* expression increased by day 7 and remained high in later time points. In contrast, macrophages from patients developing severe BPD had lower and more variable *MRC1* expression. To better understand the importance of this time dependent expression pattern, we next identified other genes in the dataset whose expression increased in resilient patients but not in patients developing severe BPD. Figure 4 shows genes of interest with patterns similar to *MRC1*. The additional macrophage scavenger receptors *MARCO* and *MSR1* also increased in macrophages from patients with no or mild BPD between day 1 and day 7 (Fig. 4b, c). Other notable genes with this similar pattern included *FABP4, C1QA, C1QB, AQP3*, and *CCL13* (Fig. 4d–h). Interestingly, *AQP3* expression (Fig. 4g) appeared to increase more gradually over time in resilient patients. These data suggested that macrophages from patients resilient to BPD dynamically expressed of a unique set of genes compared to patients with severe disease. Several of the genes displaying this pattern of expression have been reported in mature alveolar macrophages and associated with anti-inflammatory or trophic macrophage phenotypes (i.e., M2-like)[31]. In mice, alveolar macrophage differentiation occurs postnatally, with newborn lungs containing immature lung macrophages. Our data suggest a similar pattern may occur in human preterm lungs, but potentially with different transcriptional kinetics in preterm patients developing severe lung disease.

To better understand the dynamics of gene expression during normal human macrophage differentiation and development, we used a Boolean approach to analyze time-dependent changes in gene expression. StepMiner analysis[32] identified changes in macrophage gene expression over time in both severe BPD and none/mild BPD patient populations (Fig. 5a–c, Supplemental Data 6). When the genes identified in macrophages from patients with no or mild BPD were analyzed in severe BPD patients, time-dependent changes in expression were not observed (Fig. 5d). More genes were dynamically upregulated in macrophages from resilient patients (none/mild BPD; 3,654) compared to those from severe BPD patients (187) (Fig. 5e, Supplementary Data 6). Reactome analysis of genes increasing in none/mild BPD patient samples identified enrichment in mitosis/cell division, antigen presentation, and mitochondrial respiratory metabolism (Fig. 5f, Supplementary Data 7). In severe BPD patients, categories of macrophage genes increasing over time included lipid and lipoprotein metabolism, circadian clock genes, and chemokine signaling.

These analyses clearly delineated distinct transcriptional changes over the first weeks of life in the lung macrophages from patients with divergent disease courses. To gain insight into the molecular pathways that might regulate these differences, we examined the gene lists identified by StepMiner for over-represented transcription factor binding motifs. Using oPOSSUM 3.0[33], genes increasing between day 1 and day 7 in patients with none or mild BPD were more likely to contain motifs predicted to bind NKX2-5, ARID3A, HOXA5, PDX1, NKX3-1, and FOXD3 (Fig. 5g). In contrast, genes from patients developing severe BPD contained consensus motifs for binding MYC/MAX, SRF, NFKB1, MYCN, and CTCF (Fig. 5h). These differential motif enrichments point to very different transcriptional mechanisms

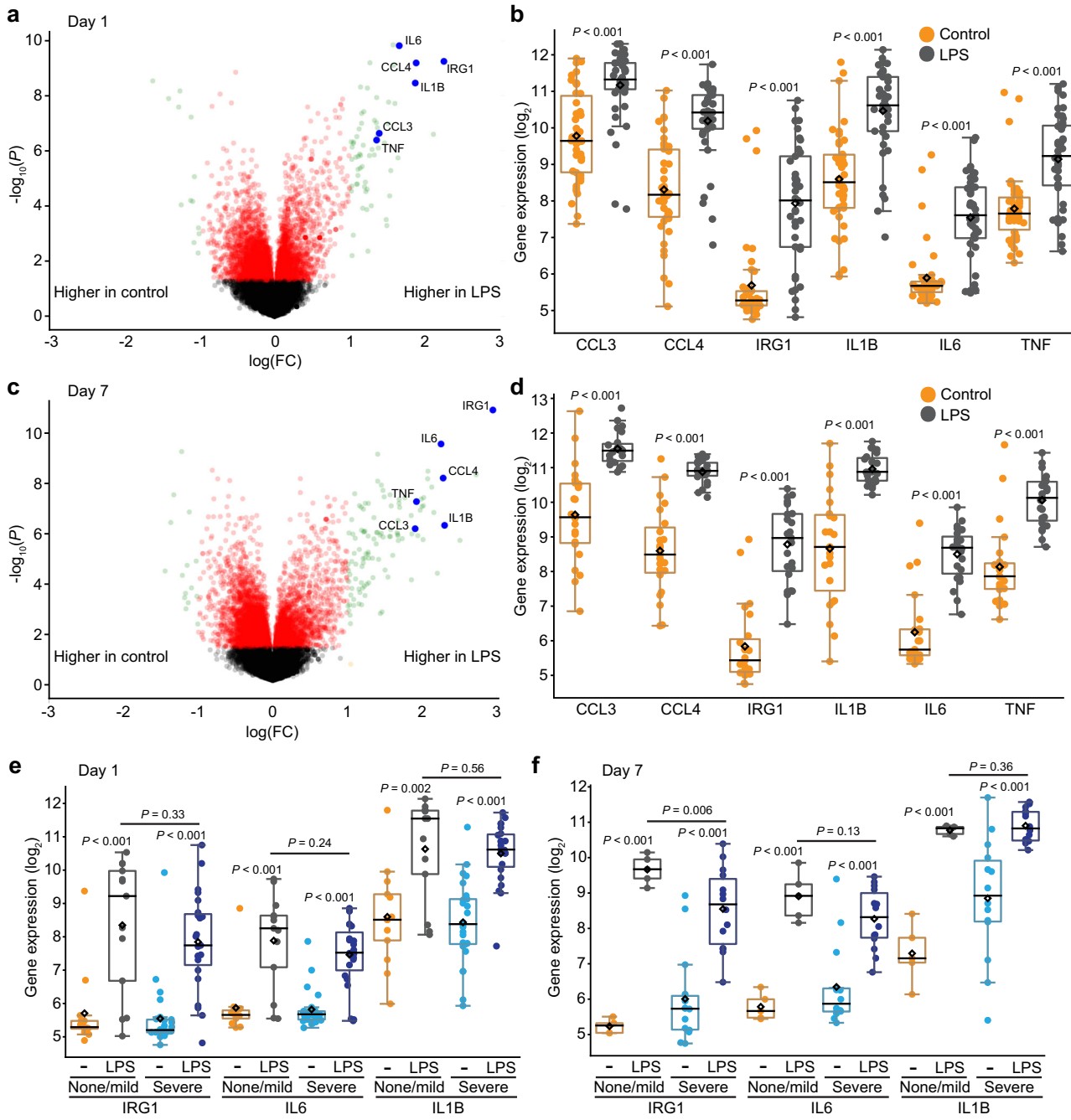

**Fig. 2 Innate immune function in preterm lung macrophages. a–d** Differential gene expression in tracheal aspirate macrophages following activation with *E. coli* LPS (250 ng/ml) for 4 h. **a** Volcano plot of day 1 samples ($n = 42$). **b** Boxplots showing data for LPS-stimulated genes *CCL3, CCL4, IRG1, IL1B, IL6,* and *TNF*. **c** and **d** LPS response in samples obtained from patients at day 7 of life, including expression of selected LPS-stimulated genes ($n = 23$). **e** and **f** LPS-stimulated expression of *IRG1, IL6,* and *IL1B* in samples obtained from patients with severe BPD vs. none/mild BPD at day 1 (**e**) and day 7 (**f**) of life ($n = 13$ for day 1 none/mild BPD; $n = 25$ for day 1 severe BPD; $n = 5$ for day 7 none/mild BPD; $n = 14$ for day 7 severe BPD). *P* values calculated by Welch Two Sample *t*-test (unpaired, two-tailed).

in patients resilient to BPD compared to those developing severe disease.

## Discussion
Here we have used transcriptional profiling to detail the molecular changes in preterm human lung macrophages in both health and disease. Compared to datasets obtained from cell lines, monocyte-derived cultured macrophages, and adult alveolar macrophages, the data presented here captured both diversity among individual patients and the unique features within the preterm lung microenvironment. Resident within the airway, tracheal aspirate macrophages are currently the most relevant cell type to BPD pathogenesis that can be obtained as part of routine neonatal intensive care. This unique dataset revealed intriguing connections between macrophage gene expression early in life and eventual BPD outcome and severity. Contrary to the notion that preterm infants have a deficient or immature innate immune response, lung macrophages from the preterm patients in our

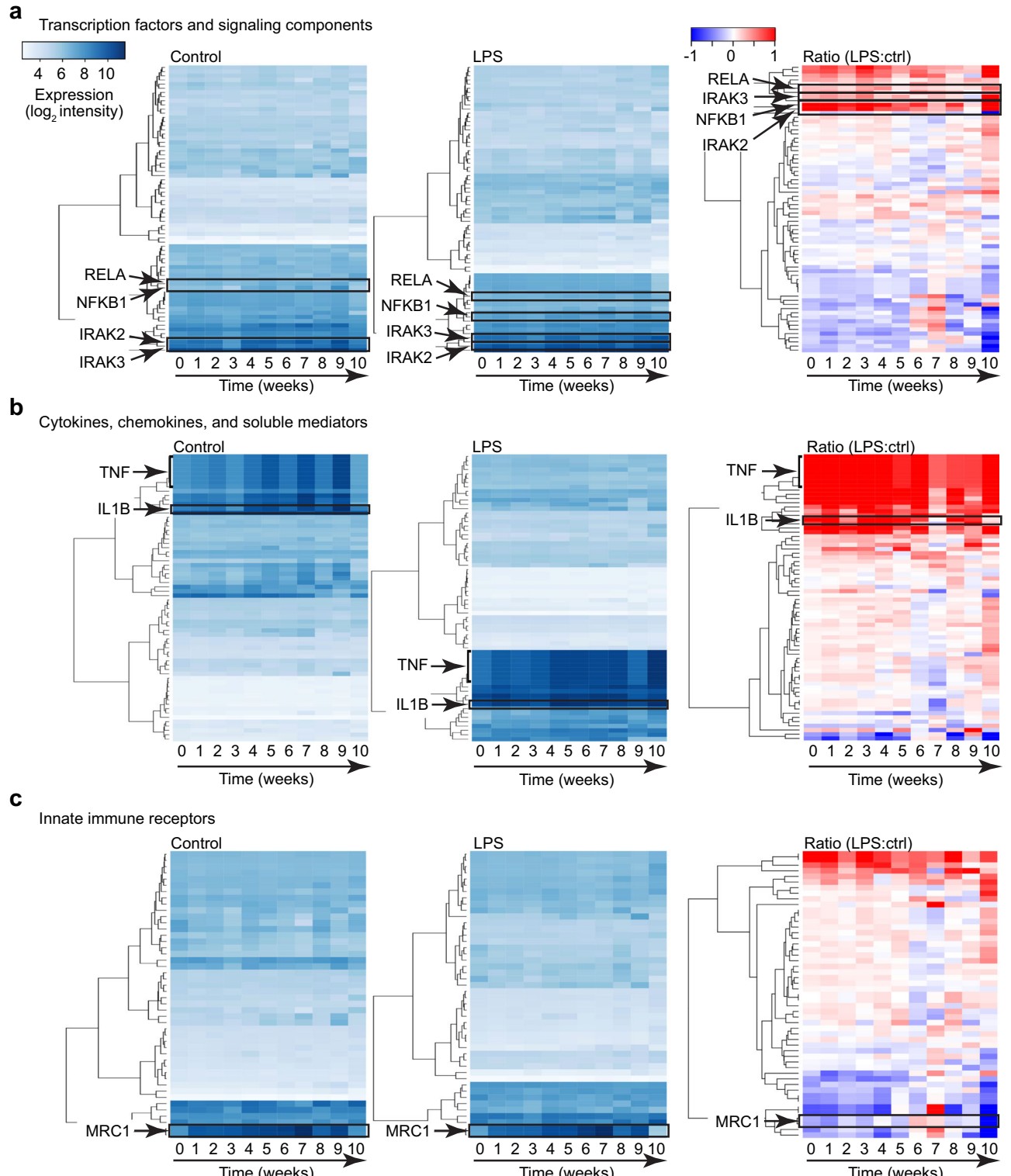

**Fig. 3 Expression of macrophage innate immune components.** Expression of innate immune transcription factors and signaling components (**a**), cytokines, chemokines, and soluble inflammatory mediators (**b**), and innate immune receptors (**c**) were profiled against time (corresponding to chronological age). Data from control samples and LPS-treated samples are shown, along with the LPS:control ratio (LPS:ctrl). Heatmaps were generated by unsupervised hierarchical clustering. Gene lists are included in Supplemental Data 5.

study mounted a robust response when challenged ex vivo with LPS. The inflammatory response was relatively consistent over in the first several weeks of life. Interestingly, we also identified a module of genes whose expression changed during the first weeks of life in patients resilient to severe BPD. These changes could

represent important molecular events in both human lung macrophage development and BPD pathogenesis.

These data strongly support the role of inflammation in BPD pathogenesis. While BPD is clearly more common in the smallest and most premature patients, maternal chorioamnionitis and

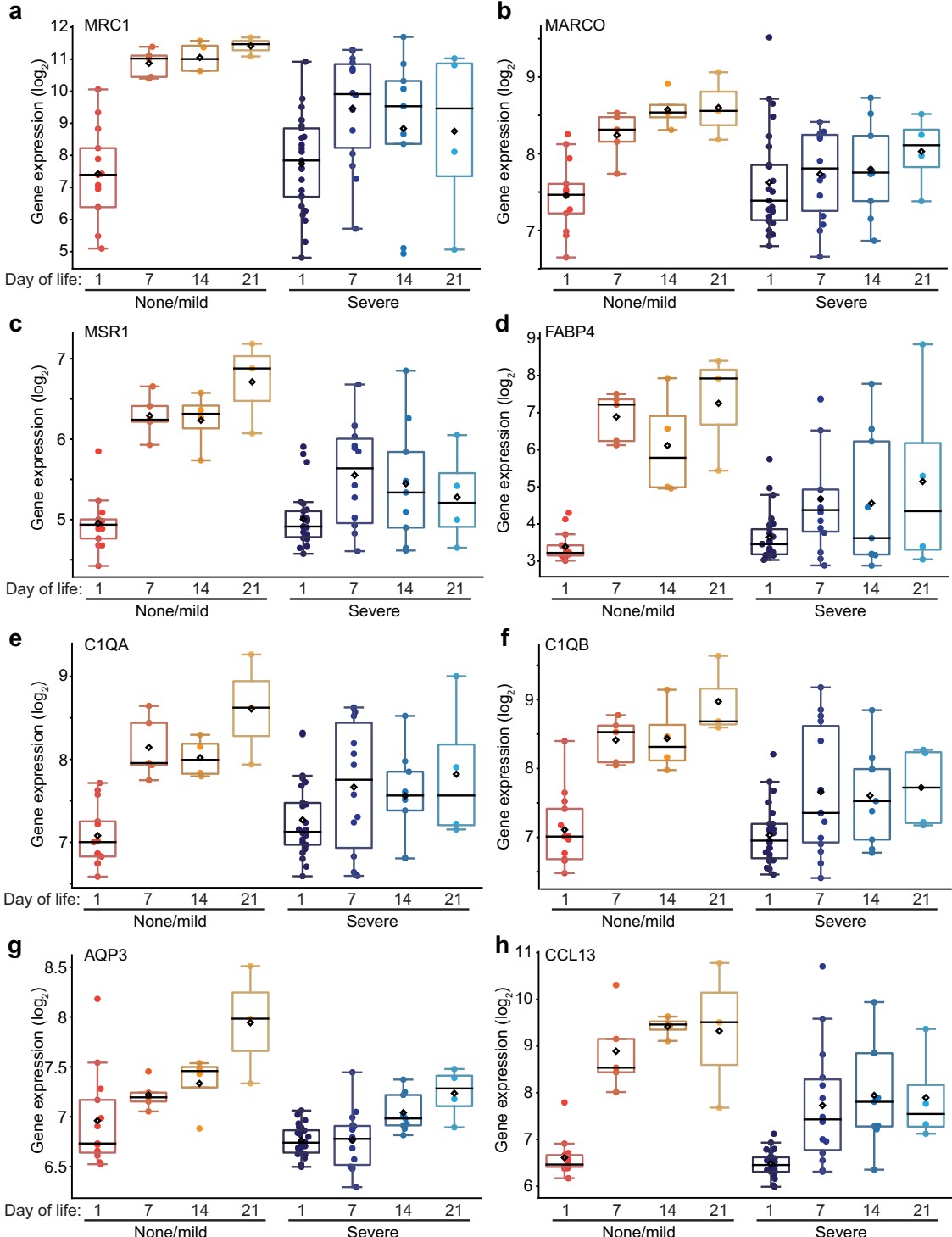

**Fig. 4 Transcriptional markers of disease resilience.** Macrophage gene expression over time plotted for samples obtained on days of life 1, 7, 14, and 21 from patients resilient to BPD (no BPD or mild BPD) and those developing severe BPD. **a** Increased expression of *MRC1* after day 1 of life in patients either not developing BPD or only developing mild disease. **b–h** Genes with similar time-dependent patterns of expression in resilient patients compared to severe BPD patients. Genes were identified by differential expression comparison between day 1 and later time points, as well as using the "Correlation" function within Hegemon. (For severe BPD, $n = 25$ on day 1, $n = 14$ on day 7, $n = 9$ on day 14, $n = 4$ on day 21; for none/mild BPD, $n = 13$ on day 1, $n = 5$ on day 7, $n = 4$ on day 14, $n = 3$ on day 21).

neonatal sepsis both increase BPD risk[34,35]. Patients developing BPD can contain elevated cytokine and chemokine levels in their blood and tracheal aspirate fluid[19,36], supporting the connections between inflammation and BPD. However, the cellular and

molecular mechanisms linking inflammation and the abnormal lung development in BPD patients remain unclear. Experimental mouse models identified lung macrophage activation as both required and sufficient to disrupt normal lung developmental

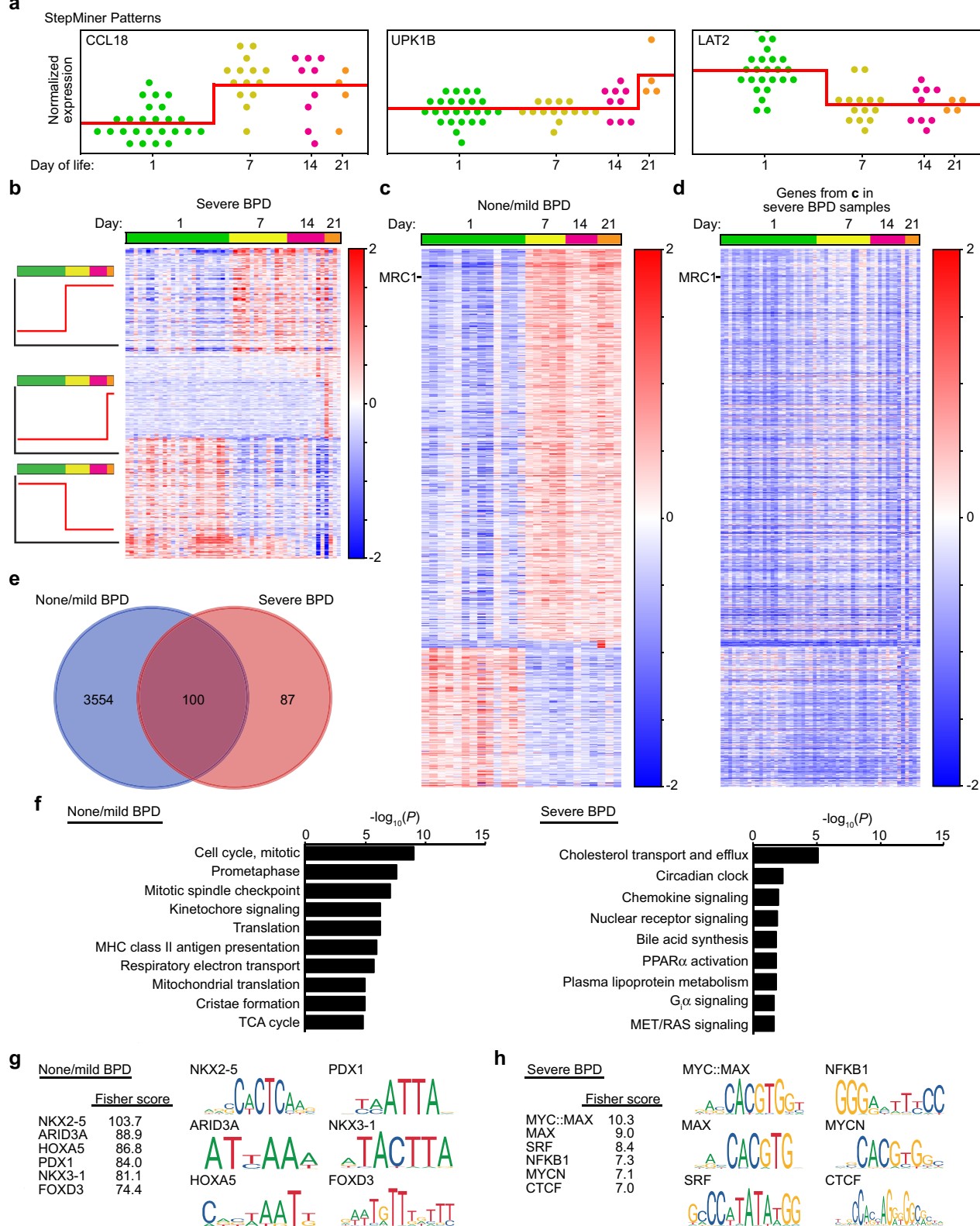

programs[25–27]. In this study, we have now demonstrated that inflammatory activation in human macrophages occurs early in disease pathogenesis. In the preterm infants studied here, lung macrophages from patients developing BPD expressed elevated levels of inflammatory chemokines and cytokines, including *CCL3* and *IL1B*. Inflammasome-mediated IL-1β release inhibits

saccular airway morphogenesis and alveolarization in mouse models[25,37,38]. Multiple lines of investigation in experimental models implicate IL-1α/β in abnormal lung development following neonatal lung injury[39,40]. Increased *Ccl3* expression in developing mice disrupted normal alveolar capillary development, likely consistent with BPD pathology[41]. Interestingly,

**Fig. 5 Temporal dynamics of gene expression in patients resilient to BPD or with severe disease. a** StepMiner analysis was used to identify distinct Boolean patterns of gene expression in patient macrophages over the first 21 days of life. Each point represents a distinct control (not LPS treated) patient sample. Analysis results shown for normalized expression of genes representative of distinct patterns in severe BPD patients. **b** and **c** Heat map analysis of the genes displaying time-dependent changes in patients with severe BPD (**b**) or patients with none or mild BPD (**c**). **d** To illustrate the unique temporal pattern of expression in patients with no BPD or mild BPD, expression of genes from **c** were plotted for patients with severe BPD. **e** Venn diagram showing that only 100 genes out of 3654 identified in patients with none or mild BPD also had time dependent expression changes in severe BPD patients. **f** Reactome analysis of time-dependent genes from **b, c. g, h** Transcription factor motif analysis of genes enriched in patients with none/mild BPD compared to patients with severe BPD. Gene lists generated from StepMiner analysis of control samples from patients with none/mild BPD (**g**) and severe BPD (**h**) were analyzed for over represented motifs using oPOSSUM 3.0. Top six motifs based on Fisher score are shown for each group along with graphical representation of their consensus motif. (For severe BPD, $n = 25$ on day 1, $n = 14$ on day 7, $n = 9$ on day 14, $n = 4$ on day 21; for none/mild BPD, $n = 13$ on day 1, $n = 5$ on day 7, $n = 4$ on day 14, $n = 3$ on day 21).

patients developing BPD expressed the inflammatory mediators *CXCL5*, *CXCL6*, *CCL3*, and *CCL20* on day 1, prior to expressing elevated *IL1B* levels. We speculate that an early inflammatory response may include select chemokines before expression of *IL1B* and *TNF*. These findings suggest a potential multi-hit hypothesis. A preterm lung with previously activated or primed chemokine-expressing macrophages at birth may be more likely to subsequently express higher levels of *IL1B*, *TNF*, or additional mediators when stimulated with a second wave of TLR and/or inflammasome activators.

The innate immune response in preterm macrophages was surprisingly stable over time. When challenged ex vivo with the TLR4 ligand LPS, lung macrophages from preterm infants expressed high levels of classic inflammatory response genes. While we did not directly compare our samples with cells from older children or adults, the LPS response in preterm infant macrophages was clearly intact at birth with minimal change over the first several weeks of life. In addition, expression of most innate immune receptors, signaling components, transcription factors, and soluble inflammatory mediators was notably consistent over time. The few exceptions included the increase in basal expression of inflammatory mediators *IL1B* and *TNF* (particularly in patients with severe BPD) and an increase in macrophage scavenger receptors. The overall immune system of preterm infants may persist in a relatively immunocompromised or tolerant state following birth, contributing to susceptibility to infection[8]. However, our data demonstrate a robust innate immune response in preterm lung macrophages.

Patients resilient to disease expressed higher levels of *C1QA/B* and the macrophage scavenger receptors *MARCO*, *MSR1*, and *MRC1*. Expression of scavenger receptors and C1Q correlates with anti-inflammatory or M2 phenotypes in other macrophage studies[31,42]. However, lung alveolar macrophages do not display a clear M1/M2 functional paradigm. We have previously demonstrated that in mice, differentiating alveolar macrophages acquire both scavenger receptor and inducible inflammatory gene expression[43]. Our data further illustrate the unique aspects of lung macrophages, as cells from patients at least one week of age and expressing these "M2-like" genes still mounted a robust response to LPS. The gene expression dynamics in preterm infants resilient to disease could represent the normal developmental process of human lung macrophage development, previously studied primarily in mouse models. In patients developing severe BPD, genes associated with alveolar macrophage development remained low, identifying a potential mechanism of disease.

Our study has several caveats. The cellular yield from endotracheal tube suctioning is quite variable, resulting in missing time points within our dataset. Cells from the tracheal aspirates were plated onto cell culture treated plastic, allowing selection of rapidly adherent macrophages above other cell populations[44,45].

Other cell types such as neutrophils were unlikely to survive and remain in culture and epithelial cells were not likely to be included in our samples due to their relatively poor adhesion properties. Importantly, while we did observe transcriptional diversity between samples, expression of key macrophage signature genes and innate immune components was consistent across samples. These data suggested that diversity was not likely to result from additional, non-myeloid cell populations. Culturing macrophages ex vivo even briefly could alter their transcriptional program[46]. However, our protocol limiting the culture duration to 4 h ex vivo was designed to minimize the potential impact of removing the cells from the lung microenvironment. With all its caveats, our tracheal aspirate isolation approach was designed for maximal feasibility in the clinical setting and reproducibility in future studies. We do anticipate that advances in cell isolation and single cell analysis will increase the resolution of various cell populations within clinical samples.

Our dataset provides among the most comprehensive patient-based analysis on the molecular pathogenesis of BPD in preterm infants. Importantly, our data link inflammation at the time of birth with the development of BPD. Other studies have measured higher levels of tracheal aspirate inflammatory mediators in patients developing BPD[47–50]. The variability between which mediators are predictive of disease progression may be related to timing of sample collection and inherent clinical variability. Based on the differential transcriptional profiles on day 1 and day 7, we speculate that during BPD pathogenesis, early expression of pro-inflammatory chemokine genes such as *CXCL5/6* and *CCL3* expands over time to include additional mediators including *IL1A* and *IL1B*. These data suggest the first week of life may mark a therapeutic window of opportunity to suppress ongoing or amplified macrophage activation and inflammation. Further high-resolution temporal studies may help define more specific molecular targets. In addition, we have identified a unique pattern of macrophage gene expression in resilient patients. Preterm infants that do not develop BPD or only develop mild disease could display normal macrophage development in the first week of life, even while receiving mechanical ventilation. Alternatively, resilient patients might instead express a protective module of transcripts that allow ongoing lung development in the face of potential injury. While obtaining lung macrophage samples over time from "normal" neonates represents a true challenge, we hope that future patient-based studies will help define the molecular mechanisms of macrophage maturation and protection against injury-mediated disruption of human lung development in vulnerable populations.

## Methods

**Study design**. This study was approved by the Institutional Review Boards at Vanderbilt University and the University of California, San Diego and Rady Children's Hospital, San Diego. Informed consent was obtained from a parent of

each subject. Tracheal aspirate samples were generated as part of routine respiratory care from patients born before 30 weeks gestation. All patients were intubated for mechanical ventilation due to respiratory distress syndrome. The initial sample for each patient was obtained within the first 24 h of life and subsequent samples were obtained weekly beginning on day 7 and continuing weekly if the patient remained intubated. Some of the participating infants at Vanderbilt were enrolled in the Prematurity and Respiratory Outcome Program (PROP) study at Vanderbilt University Medical Center (NCT01460576)[51]. BPD severity was assigned according to National Institutes of Health guidelines[28]. All patients requiring oxygen for 28 days were diagnosed with BPD. Patients with mild BPD were in room air by 36 weeks corrected age. Patients with moderate BPD were receiving supplemental fraction of inspired oxygen of <0.30 at 36 weeks corrected age. Severe BPD was diagnosed if patients were receiving >0.30 fraction of inspired oxygen or any positive pressure support at 36 weeks corrected age.

Tracheal aspirate cells were collected from intubated preterm infants as detailed below. After instillation of 0.5 ml of normal saline into the endotracheal tube, a suction catheter was introduced into the endotracheal tube and tracheal secretions were aspirated into a sterile mucus trap. The suction catheter was subsequently rinsed with an additional 1.5 ml of sterile saline to clear any remaining aspirate from the catheter into the mucus trap. Cells were collected by centrifugation and resuspended in DMEM with 10% fetal bovine serum, penicillin, and streptomycin. Equal aliquots of resuspended cells were divided between two separate tissue culture treated plates and incubated at 37 °C in a humidified environment of 95% air and 5% $CO_2$ to allow macrophage attachment. After 30 min, nonadherent cells were removed by gentle washing and fresh cell culture media was applied. Lipopolysaccharide from *E. coli* (strain 055:B5, gel purified, Sigma-Aldrich L2637; 250 ng/ml) was added to one of the plates and both were incubated for 4 h at 37 °C. After incubation, the media was removed and cells were lysed in TRIzol monophasic solution containing phenol and guanidine isothiocyanate (Thermo Fisher). RNA was isolated using standard protocols for TRIzol extraction. RNA yield and quality were measured using High Sensitivity RNA ScreenTape Analysis (Agilent Tapestation).

**Human transcriptome arrays 2.0 (HTA) microarray analysis**. For transcriptional profiling using the Affymetrix GeneChip Human Transcriptome 2.0 Arrays, 1 ng of total RNA was processed using the WT Pico Kit. Fragmentation, hybridization, and scanning were all performed according to Affymetrix manufacturer's instructions. Raw expression values were extracted from Affymetrix CEL-files using *affy* R/Bioconductor package (version 1.52.0)[52]. Background adjustment and quantile normalization were performed using the Robust Multichip Average algorithm in oligo R/Bioconductor package (version 1.38.0)[53]. Probe annotations were obtained by matching the probe IDs to the manufacturer's provided data files ("HTA-2 0.na35.2.hg19.transcript.csv" and "HTA-2 0.na35.2.hg19.probeset.csv").

Pseudotime analyses were performed by grouping the normalized expression values by gestational age using *aggregate/stats* (function = mean) R package. The mean values were used to calculate the gene expression ratio or fold-change (LPS:ctrl). The normalized expression values and fold-change were used for heatmaps, which were generated using heatmap.2 function from *gplots* R package version 3.8.1. Genes were clustered using unsupervised hierarchical clustering with Euclidean distance.

**Pathway analysis**. Differentially expressed gene lists were analyzed via the Reactome database and algorithm[29]. Reactome identifies signaling and metabolic molecules and organizes their relations into biological pathways and processes. Lists of significantly represented pathways and processes were generated based on decreasing $-\log_{10}(P)$, with the P values obtained from over-representation analysis. The lists of genes comprising each pathway are available at https://reactome.org/PathwayBrowser/.

**StepMiner analysis**. StepMiner is a computational tool that identifies step-wise transitions in a timeseries data[32]. StepMiner performed an adaptive regression scheme to identify the best possible step up or down based on sum-of-square errors. The steps were placed between time points at the sharpest change between low expression and high expression levels, identifying timepoints of the gene expression-switching event. To fit a step function, the algorithm evaluated all possible step positions, and for each position, computed the average of the values on both side of the step for the constant segments. An adaptive regression scheme was used to choose the step positions that minimize the square error with the fitted data. Finally, a regression test statistic was computed as follows:

$$F \text{ stat} = \frac{\sum_{i=1}^{n}(\widehat{X_i} - \bar{X})^2/(m-1)}{\sum_{i=1}^{n}(X_i - \widehat{X_i})^2/(n-m)}$$

Where $X_i$ for $i = 1$ to $n$ are the values, $\widehat{X_i}$ for $i = 1$ to $n$ are fitted values. $m$ is the degrees of freedom used for the adaptive regression analysis. $\bar{X}$ is average of all the values. StepMiner analysis was used with a P-value threshold of 0.01 along the time course data for day 1, 7, 14, and 21 in patient samples from both none/mild and severe BPD cases separately.

**Hegemon data visualization platform**. The normalized data were uploaded to our Hegemon (Hierarchical exploration of gene expression microarray online) data

analysis platform[54–56]. This website provides several tools to explore gene expression datasets. For the data presented here, Hegemon was used to identify differentially expressed gene lists, boxplots and statistical comparison of gene expression between multiple patient samples, identification of genes with high positive or negative correlation of expression with select target genes, and Boolean implication relationships between pairs of genes[57].

**Transcription factor motif analysis**. oPOSSUM 3.0 (opossum.cisreg.ca/oPOSSUM3) was used to identify over-represented conserved transcription factor binding sites within the sets of genes generated by StepMiner analysis. Human Single Site Analysis used Ensembl release 64 and all 24,752 genes in the oPOSSUM database as background genes. All vertebrate JASPAR CORE transcription factor profiles were tested. An 85% matrix score threshold and 0.4 conservation cutoff were used to search sequences +/− 5000 bp from the transcriptional start sites of each gene. Results were reported by Fisher score, using a cutoff value of 7. Top scoring motifs are presented in the Results.

**Statistics and reproducibility**. For differential expression analysis, standard t-tests were performed using python scipy.stats.ttest_ind package (version 0.19.0) with Welch's Two Sample t-test (unpaired, unequal variance (equal_var = False), and unequal sample size) parameters. Multiple hypothesis corrections were performed by adjusting P values with statsmodels.stats.multitest.multipletests (fdr_bh: Benjamini/Hochberg principles). The results were independently validated with R statistical software (R version 3.6.1; 2019-07-05). Volcano plots were prepared using python matplotlib (version: 2.1.1). Boxplots were generated using matplotlib.pyplot.boxplot and beeswarm package. A diamond sign in the boxplot indicates the mean (average) expression value. Real time PCR was performed in triplicate for each sample using GAPDH as a control. Statistical analysis was performed by comparing the $\Delta C_T$ values by Mann–Whitney U test.

**Reporting summary**. Further information on research design is available in the Nature Research Reporting Summary linked to this article.

## Data availability

All microarray data are publicly available through the Gene Expression Omnibus (GSE149490). Source data for Fig. 1 are in Supplementary Data 8.

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

## Acknowledgements

We would like to thank Lisa Lepis, Renee Bridge, Ellen Knodel, Steven Steele, Amy Law, Mamon Dey, and Amanda Im for their assistance with this project. We also thank Lori Broderick, Hal Hoffman, Mamata Sivagnanam, and Eniko Sajti for their helpful advice. This project would not be possible without the help and support from the NICU staff at Vanderbilt Children's Hospital, University of California, San Diego, and Rady Children's Hospital. This work was funded by the NHLBI (HL126703, HL097195, HL101456), Gerber Foundation (1823-3830, 20180324), the UC San Diego Clinical and Translational Research Institute (CIIPLSP), and the Rady Children's Hospital Academic Enrichment Fund (1001328-AWD CR30).

## Author contributions

D.S., U.M., S.T., D.D., L.S.Z., K.Z., designed, conducted, and interpreted the bioinformatic analysis of the microarray dataset. G.E.H, A.N.S. R.M.M., A.M.M. isolated tracheal aspirate cells, extracted RNA and prepared samples for transcriptional profiling. G.E.H. processed samples and performed microarray hybridization and scanning. J.L.A., T.S.B., and L.S.P. designed the study and supervised its performance. D.S., L.S.Z., and L.S.P. wrote the initial drafts of the manuscript.

## Competing interests

The authors declare no competing interests.
