## [Peer Review File · Communications Biology]

Reviewers' comments:

Reviewer #1 (Remarks to the Author):

In their study "Transcriptional profiling of lung macrophages from preterm infants identifies disease related programs", Sahoo et al. perform large scale microarray-based gene expression analysis of lung tracheal aspirate samples obtained from pre-term infants. The authors describe cellular gene configurations at steady state and upon LPS stimulation at different time points after birth.

Overall, the study is interesting, informative and quite well written. However, the text language of the study could be improved and spelling and grammar mistakes should be corrected. The data presented in the study are novel and interesting but are often not optimally displayed and lack information. As this study is rather descriptive, it would be important to provide additional information, for instance supplementary tables comprising gene lists and numbers of genes (at least for significantly up and down-regulated genes). The statistical techniques are often insufficiently described (e.g. no specifications of cut-offs, no indications of FDR correction) or suboptimal for the study design. For future submissions, please include page numbers and line numbers in the manuscript in to allow more efficient referencing.

Altogether, this study would need a few major, but feasible improvements in the presentation and statistical analysis of the data. However this study would deserve to be published in Communications Biology as it provides valuable insights into the transcriptional regulation of human alveolar macrophages in the context of BPD in preterm infants.

Comments and questions:

General:

In general, this study could be greatly improved by providing details about the patients used for the different experiments/in different Figures. In order to understand the influence of confounding parameters affecting BPD development, a table comprising the gestational stage, sex etc. of individual patients should be included in the supplements, indicating also which patients/time points were analyzed in which figure.

The authors should also provide (supplementary) tables of gene expression data for the different analyses in order to support rather less informative graphs, like heat maps and volcano plots.

The resolution of heat map pictures is rather bad quality (at least in the paper proof) and should be improved.

The figure legend text should be purely descriptive of what is displayed in the figure and not comment on the results.

Specific:

Figure 1

1. Please include the individual data points of each replicate within the boxplots.
2. The volcano plots – as currently displayed - do not provide useful information. Please increase the size of the volcano plots and highlight the genes shown in the boxplots. It would also be helpful to state the number of genes differentially expressed and provide detailed gene lists in the supplements.
3. According to the supplementary table, there should be a higher number of patients in the "no BPD" and "BPD" groups than stated in the figure legend. Please specify the reasons for not including all day 1 patient samples.

In the main text, the latest time point of sampling is stated to be 23 weeks of chronological age, but in the corresponding Suppl. Fig. S1, a 24 week time point is shown as the sampling time point.

4. It is unclear whether the patients analyzed on day 1 and day 7 are the same or not. Again, further information about the patients could resolve this issue.

5. In figure 1a-c "No BPD" is compared with "BPD", however, in figure 1d-i, the comparison is made between "None/Mild BPD" and "Severe BPD". The indicated reason for combining "None" and "Mild" is

understandable, however, to make the overall analysis more consistent, these groups should be combined consistently throughout the whole study (i.e. also in Figure 1a-c).

6. What is the basis for selecting the genes shown in 1f-i for individual display. Are these the ones that are significantly differentially expressed? Or are they the most "interesting" ones because they are proinflammatory? In any case, IL-10 signaling, which is the top hit in 1f, is generally considered anti-inflammatory. This should be mentioned and discussed in the text.

7. It is not known which specific genes constitute the pathway groups shown in Figure 1f-i. One way to show this would be as part of gene expression tables or individual supplementary tables.

8. The authors appear not to do Multiple testing corrections which might substantially affect the number of differentially expressed genes. More appropriate tools for microarray statistical analysis (e.g LIMMA) should be used instead of t-tests.

9. In Figure 1a, 1b, 1d and 1e, there is no unit for the gene expression on the y axis. Is it a fold change or an absolute value?

Figure 2:

1. What is the rationale behind using an LPS concentration of 250ng/ml. This appears to be a very high concentration and could mask potential differences between the compared groups. Was a dose response experiment performed to identify the ideal LPS concentration? If so, please provide the data. Why are the cells stimulated for particularly 4 hours?

2. It would be important to report the purity of the tracheal aspirates before and after seeding at least exemplarily. Flow cytometry data would be convincing.

3. In figure 2c and 2d, statistics should additionally be provided for the "None/Mild" vs "Severe" LPS-stimulated groups.

4. A tracheal aspirate volume of 500 μ l might allow measurement of inflammatory mediators, for instance by ELISA. This could provide interesting additional information. Were (or could) such analyses (be) performed?

5. Comments regarding the patient information, the representation of the data as well as the statistical analysis made in Figure 1 also apply here.

Figure 3:

1. The figure title states that the infants are "extremely" preterm. What are the criteria for this statement?

2. Heatmaps should be provided with higher resolution.

3. In figure 3a-c, the genes highlighted in the control group (e.g. TNF and IL1B) are not highlighted in the LPS group. They should be highlighted in both groups to enable proper comparisons.

Do the "Ratio" heatmaps include the same genes as in the "control" and "LPS" groups? A list of genes would be again of great help here.

4. What are the individual columns of the heatmaps representing? Which exact timepoints are being analyzed? How many patients from which time point? In the current form, the heatmaps are not informative enough.

5. In Figure 3a-c, the color key for "control" vs "LPS" panels would be much clearer to understand if displayed as one color increasing monotonously. The way they are currently displayed, the reader easily believes that blue means downregulated, red means upregulated. The colors for the ratios don't need to be changed.

6. Instead of showing heatmaps, an alternative would be to show the expression of the indicated genes over time. However, if heatmaps are displayed properly with all necessary information, this might not be necessary.

Figure 4:

1. It is unclear whether the samples compared in "None/Mild" and "Severe" groups derive from the same or different patients throughout the analysis.

2. Again, please show the individual data points within the box plots.
3. More detailed information regarding which patients were included at which time points would be helpful to interpret the data. Would it be possible to restrict the analysis to the samples that are present in all time points?
4. Please state the criteria for identification of genes that correlate with MRC1 expression, the correlation test is not stated (As currently displayed, the pattern looks similar by eye only). Why are these correlated genes not found in the heatmaps of Figure 3 if they behave similar to MRC1?
5. Please provide statistic to support the claims that "None/Mild" and "Severe" groups are different.
6. Previously stated comments regarding the figure legend description also apply here.

Figure 5 :

1. Do the symbols in 5a represent individual patients?
2. In figure 5b, the upper separation line on the left side seems to be misplaced.
3. In figure 5g and 5h, the analyzed time points are unclear.
4. Sample numbers are missing for the whole figure.

Supplementary Figure S2:

The resolution of these heatmaps is too low. Please provide additional information regarding the data sets (e.g. how the samples were collected, whether they were obtained from healthy patients, etc).

Supplementary Figure S3:

What does this Figure add to the story? It would be important to provide information reg. its value and instructions on how to read or interpret this plot. Also, the color scale is hard to discriminate. And what does it show?

Methods section:

Overall, the methods are fine but please provide more detailed information reg. reagents, e.g. LPS (supplier and order number) and Trizol. The paragraph about the Hedgemon tool is described in too much detail and with inappropriate wording. Please reduce/rephrase this description to make it sound less "advertising". Please specify for which analysis (Figures) it was used.

Reviewer #2 (Remarks to the Author):

This manuscript investigates the role of macrophages in bronchopulmonary dysplasia (BPD). To accomplish this, transcriptome analysis using Affymetrix microarrays was used to carry out an unbiased survey of the differences in gene expression profiles in tracheal aspirate macrophages collected from intubated preterm infants. In this study, the authors show that macrophages isolated on the first day of life from patients who go on to develop BPD express higher levels of inflammatory genes such as CCL3, CCL4, CCL20, and IL23. Some gene changes were validated using quantitative real time PCR. At day 7 of age, preterm infants that eventually developed severe BPD expressed higher levels of macrophage inflammatory mediators such as IL1A and IL1B. Pathway analysis revealed that inflammatory cytokine signaling pathways were highly represented in patients developing BPD. The authors also showed that macrophages from patients with no or mild BPD expressed a unique set of genes in the first week of life that are M2-like (high MRC1 and C1Q). Pathway analysis of genes that changed between days 1 and 7 revealed enrichment for genes involved with antigen presentation and mitochondrial respiratory metabolism in none/mild BPD patients, whereas in severe BPD patients, genes that changed were associated with chemokine signaling and lipid/lipoprotein metabolism. These data strongly support the role of macrophages and inflammation in the pathogenesis of BPD.

Overall, the manuscript is well written. The experiments performed are logical and the authors' conclusions are supported by their results. The study is by nature descriptive, and how these findings can be applied to the treatment of preterm infants at risk for BPD is unclear. However, the authors' findings are significant. Regarding the manuscript, I have no major concerns and only a couple minor concerns:

Major concerns: None

Minor concerns:

1. Figs 1f and 1h: Reactome analysis shows that genes in the IL-10 signaling pathway are higher in BPD at day 1 and in severe BPD at day 7. However, IL-10 levels have been reported to be decreased in tracheal aspirates and lung tissue sections from preterms who develop BPD (Oei et al, *Acta Pediatr* 2002; McColm et al, *Arch Dis Child Fetal Neonatal Ed* 2000; Garingo et al, *Pediatr Res* 2007). Although enrichment for genes in IL-10 signaling does not indicate that preterms who developed BPD had higher expression of IL-10, the authors may want to comment on this potential discrepancy.
2. Previously published studies have found elevated levels of multiple proinflammatory cytokines and chemokines in tracheal aspirates from preterm infants with BPD, including IL-1B, IL-6, IL-8, IL-16, TNF α , IFN γ , CCL2, CCL7, and CCL8. Although the authors show that transcriptional profiling confirms elevated levels of IL-1B, other cytokines and chemokines that have been reported were not found in this study. Again, the authors may want to comment on differences between other published studies and their findings.

We appreciate the reviewers' overall enthusiasm for the novelty and importance of our findings. In this revised manuscript, we have responded to each and every comment and suggestion supplied by the reviewers and feel that the paper is now significantly improved. Below is a detailed response to each comment.

Reviewer #1 (Remarks to the Author):

The data presented in the study are novel and interesting but ...it would be important to provide additional information, for instance supplementary tables comprising gene lists and numbers of genes (at least for significantly up and down-regulated genes).

We have included supplementary tables with gene lists and numbers along with including details in the methods section regarding the statistical analyses used to generate the lists.

For future submissions, please include page numbers and line numbers in the manuscript in to allow more efficient referencing.

We have included these.

Comments and questions:

General:

...a table comprising the gestational stage, sex etc. of individual patients should be included in the supplements, indicating also which patients/time points were analyzed in which figure.

We have included Supplemental table 2 with more detail about the samples obtained and analyzed, including the number of samples included from each patient, their gestation, weight, gender, etc.

The authors should also provide (supplementary) tables of gene expression data for the different analyses in order to support rather less informative graphs, like heat maps and volcano plots.

Additional supplementary tables are now included to support the heat maps and volcano plots. Files including our data have been deposited to GEO and will be publicly available upon publication.

The resolution of heat map pictures is rather bad quality (at least in the paper proof) and should be improved.

We apologize for the apparent loss of resolution in the heat maps. Higher resolution images are included with the resubmission.

The figure legend text should be purely descriptive of what is displayed in the figure and not comment on the results.

The figure legends have been revised.

Specific:

Figure 1

1. Please include the individual data points of each replicate within the boxplots.

New boxplots include individual data points.

2. The volcano plots – as currently displayed - do not provide useful information. Please increase the size of the volcano plots and highlight the genes shown in the boxplots. It would also be helpful to state the number of genes differentially expressed and provide detailed gene lists in the supplements.

Revised volcano plots are larger and highlight the specific genes shown in the boxplots. Supplemental table S3 includes the number of differentially expressed genes and the lists of specific genes.

3. According to the supplementary table, there should be a higher number of patients in the “no BPD” and “BPD” groups than stated in the figure legend. Please specify the reasons for not including all day 1 patient samples.

All day 1 patient samples were included. We were not able to obtain sufficient mRNA yields from every sample. Therefore, the number of samples analyzed in each group is smaller than the actual number of patients in each group. We have provided additional detail within the supplemental information (Table S2) that details the number of patients and samples in the overall study.

In the main text, the latest time point of sampling is stated to be 23 weeks of chronological age, but in the corresponding Suppl. Fig. S1, a 24 week time point is shown as the sampling time point.

The main text is correct. We have revised Supplemental Figure 1 to indicate that the last sample obtained was at 23 wk of chronological age.

4. It is unclear whether the patients analyzed on day 1 and day 7 are the same or not. Again, further information about the patients could resolve this issue.

As mentioned above in “3,” we have included additional information about patients and samples to clarify further. In regard to day 1 and day 7 samples, we obtained data from 42 patients on day 1. We also obtained day 7 samples from 12 of these patients. In addition, we obtained data on day 7 from an additional 11 patients that did not have data on day 1, giving a total of 23 patient samples on day 7.

5. In figure 1a-c “No BPD” is compared with “BPD”, however, in figure 1d-i, the comparison is made between “None/Mild BPD” and “Severe BPD”. The indicated reason for combining “None” and “Mild” is understandable, however, to make the overall analysis more consistent, these groups should be combined consistently throughout the whole study (i.e. also in Figure 1a-c).

The reviewer does make a good point. We chose to combine groups in this manner as the number of samples at day 7 is understandably lower than at day 1. In addition, we feel that understanding the unique aspects of patients still intubated at day 7 but not developing severe BPD makes this comparison compelling. We have included additional text in the revised manuscript to discuss this point.

6. What is the basis for selecting the genes shown in 1f-i for individual display. Are these the ones that are significantly differentially expressed? Or are they the most “interesting” ones because they are proinflammatory? In any case, IL-10 signaling, which is the top hit in 1f, is generally considered anti-inflammatory. This should be mentioned and discussed in the text.

All genes differentially expressed were included in Reactome analysis. The Reactome categories shown were the top categories based on statistical significance. Importantly, Reactome categories include multiple genes within various pathways, some of which may have different functional activities. Regarding IL-10 signaling, the genes mapping to that category include IL1A, IL1B, and CCL3, considered inflammatory mediators. We have included this explanation in the revised text.

7. It is not known which specific genes constitute the pathway groups shown in Figure 1f-i. One way to show this would be as part of gene expression tables or individual supplementary tables.

Supplemental tables have been included with the Reactome pathway reports. We have also included a link to the Reactome database listing the specific genes included in each pathway.

8. The authors appear not to do Multiple testing corrections which might substantially affect the number of differentially expressed genes. More appropriate tools for microarray statistical analysis (e.g LIMMA) should be used instead of t-tests.

Standard t-tests were performed using python `scipy.stats.ttest_ind` package (version 0.19.0) with Welch's Two Sample t-test (unpaired, unequal variance (`equal_var=False`), and unequal sample size) parameters. Multiple hypothesis correction were performed by adjusting p values with `statsmodels.stats.multitest.multipletests` (`fdr_bh`: Benjamini/Hochberg principles). The results were independently validated with R statistical software (R version 3.6.1; 2019-07-05). Volcano plots were prepared using python `matplotlib` (version: 2.1.1). We have included additional detail regarding these analyses in the revised manuscript.

9. In Figure 1a, 1b, 1d and 1e, there is no unit for the gene expression on the y axis. Is it a fold change or an absolute value?

Units for the y axis are log2 of the gene expression. This is included on the revised figures.

Figure 2:

1. What is the rationale behind using an LPS concentration of 250ng/ml. This appears to be a very high concentration and could mask potential differences between the compared groups. Was a dose response experiment performed to identify the ideal LPS concentration? If so, please provide the data. Why are the cells stimulated for particularly 4 hours?

We and others have published numerous manuscripts using 250 ng/ml of E. coli LPS in stimulating the lung macrophage innate immune response. The Reviewer is correct in suggesting that some immune cells will respond to lower doses of LPS. However, we did not have sufficient cell numbers to perform dose response experiments for patient samples. Measuring the transcriptional response at 4 h is also consistent with many published studies aimed at detected a maximal number early/intermediate phase, LPS-induced transcripts.

2. It would be important to report the purity of the tracheal aspirates before and after seeding at least exemplarily. Flow cytometry data would be convincing.

We did not have sufficient cell number to perform FACS on isolated cells for each sample. Our macrophage isolation protocol has been previously published for both human and animal studies (references 43 and 44) and routinely used in both the Prince and Blackwell laboratories. Our gene expression data presented are also consistent with the macrophage transcriptional signature.

3. In figure 2c and 2d, statistics should additionally be provided for the “None/Mild” vs “Severe” LPS-stimulated groups.

These data have been included.

4. A tracheal aspirate volume of 500 μ l might allow measurement of inflammatory mediators, for instance by ELISA. This could provide interesting additional information. Were (or could) such analyses (be) performed?

We agree that tracheal aspirates might contain interesting information regarding inflammatory mediator concentrations. However, this study focused on cell isolation and transcriptional profiling, so peptide concentrations were not measured.

5. Comments regarding the patient information, the representation of the data as well as the statistical analysis made in Figure 1 also apply here.

We have addressed these issues as above for Figure 1.

Figure 3:

1. The figure title states that the infants are “extremely” preterm. What are the criteria for this statement?

Infants in the study were born before 30 wk gestation. We have removed the word “extremely” to be consistent throughout the manuscript.

2. Heatmaps should be provided with higher resolution.

We apologize for the compression-mediated lack of resolution in generating the pdf. Figures with appropriate resolution have been included with our resubmission.

3. In figure 3a-c, the genes highlighted in the control group (e.g. TNF and IL1B) are not highlighted in the LPS group. They should be highlighted in both groups to enable proper comparisons.

Highlighted genes are now indicated in all heatmaps.

Do the “Ratio” heatmaps include the same genes as in the “control” and “LPS” groups? A list of genes would be again of great help here.

The same genes are included in each heatmap. The lists of genes in each set of heatmaps are now included in Supplementary Table S6.

4. What are the individual columns of the heatmaps representing? Which exact timepoints are being analyzed? How many patients from which time point? In the current form, the heatmaps are not informative enough.

The columns represent chronological age. 11 different time points are included, from week 0 (day 1) to week 10. The number of samples at each time point is shown in Supplemental figure 1.

5. In Figure 3a-c, the color key for “control” vs “LPS” panels would be much clearer to understand if displayed as one color increasing monotonously. The way they are currently displayed, the reader

easily believes that blue means downregulated, red means upregulated. The colors for the ratios don't need to be changed.

We have changed the color scheme for figure 3 according to the reviewer's suggestion.

6. Instead of showing heatmaps, an alternative would be to show the expression of the indicated genes over time. However, if heatmaps are displayed properly with all necessary information, this might not be necessary.

We appreciate the alternative idea. By incorporating the reviewer's suggestions, we believe that the revised heatmaps might be the ideal way to present the data.

Figure 4:

1. It is unclear whether the samples compared in "None/Mild" and "Severe" groups derive from the same or different patients throughout the analysis.

As we indicated earlier, only a few patients are represented at each time point. Some samples did not contain sufficient mRNA for analysis. In addition, some patients were extubated and removed from mechanical ventilation, making it impossible to obtain samples at later time points. In total, three patients (two with severe BPD and one with mild BPD) had samples represented at all four time points in Figures 4 and 5.

2. Again, please show the individual data points within the box plots.

We have included revised figures to now show each data point.

3. More detailed information regarding which patients were included at which time points would be helpful to interpret the data. Would it be possible to restrict the analysis to the samples that are present in all time points?

Additional data are included in Supplemental Table 2. We feel that restricting the analysis to only three patient samples would not permit rigorous statistical analysis.

4. Please state the criteria for identification of genes that correlate with MRC1 expression, the correlation test is not stated (As currently displayed, the pattern looks similar by eye only). Why are these correlated genes not found in the heatmaps of Figure 3 if they behave similar to MRC1?

We first measuring the changes in MRC1 expression over time in both severe BPD and none/mild BPD patients as suggested by data seen in Figure 3c. We next performed similar analysis on genes related to MRC1 (other scavenger receptors, MSR1, MARCO). Additional genes were identified by comparing gene expression at day 1 with later time points in patients resilient to severe BPD. In addition, we used the "corr" function within Hegemon to identify genes whose expression correlated with MRC1 across the entire dataset. As we identified some intriguing patterns of expression between groups, we decided to perform an unbiased Boolean analysis (StepMiner) to rigorously identify genes with time-dependent changes in expression. We have included additional text describing this approach in the Results section.

5. Please provide statistic to support the claims that "None/Mild" and "Severe" groups are different.

We have included additional detail on how we identified MRC1 and other genes presented in this figure. Statistical analysis is built into the StepMiner algorithm that identified genes with time-dependent Boolean relationships in Figure 5.

6. Previously stated comments regarding the figure legend description also apply here.

We have revised the figure legends.

Figure 5 :

1. Do the symbols in 5a represent individual patients?

Yes. Each point in Figure 5a represents an individual sample.

2. In figure 5b, the upper separation line on the left side seems to be misplaced.

The figure has been revised and corrected.

3. In figure 5g and 5h, the analyzed time points are unclear.

For Figures 5f-h, all genes identified by StepMiner as having time-dependent increases between day 1 and day 21 were included in oPOSSUM motif enrichment analysis. These genes correspond to the ones indicated in the heat maps in Figure 5b,c and the Venn diagram in Figure 5e. We have clarified this in the figure legend.

4. Sample numbers are missing for the whole figure.

The sample numbers are identical to those indicated for revised Figure 4. We have also included this information in the legend of revised Figure 5.

Supplementary Figure S2:

The resolution of these heatmaps is too low. Please provide additional information regarding the data sets (e.g. how the samples were collected, whether they were obtained from healthy patients, etc).

We apologize for the loss of resolution during pdf creation. Higher resolution figures are submitted. Details on the datasets are available at <https://www.ncbi.nlm.nih.gov/geo/query/acc.cgi?acc=GSE134312>.

Supplementary Figure S3:

What does this Figure add to the story? It would be important to provide information reg. its value and instructions on how to read or interpret this plot. Also, the color scale is hard to discriminate. And what does it show?

This figure illustrated the pathway analysis using the Reactome Pathway Browser. As the data are also represented in Figure 5f, we have removed Supplemental Figure S3.

Methods section:

Overall, the methods are fine but please provide more detailed information reg. reagents, e.g. LPS (supplier and order number) and Trizol. The paragraph about the Hedgemon tool is described in too much detail and with inappropriate wording. Please reduce/rephrase this description to make it sound less “advertising”. Please specify for which analysis (Figures) it was used.

We have included additional details in the Methods section and have edited the descriptions of analysis tools.

Reviewer #2 (Remarks to the Author):

This manuscript investigates the role of macrophages in bronchopulmonary dysplasia (BPD). To accomplish this, transcriptome analysis using Affymetrix microarrays was used to carry out an unbiased survey of the differences in gene expression profiles in tracheal aspirate macrophages collected from intubated preterm infants. In this study, the authors show that macrophages isolated on the first day of life from patients who go on to develop BPD express higher levels of inflammatory genes such as CCL3, CCL4, CCL20, and IL23. Some gene changes were validated using quantitative real time PCR. At day 7 of age, preterm infants that eventually developed severe BPD expressed higher levels of macrophage inflammatory mediators such as IL1A and IL1B. Pathway analysis revealed that inflammatory cytokine signaling pathways were highly represented in patients developing BPD. The authors also showed that macrophages from patients with no or mild BPD expressed a unique set of genes in the first week of life that are M2-like (high MRC1 and C1Q). Pathway analysis of genes that changed between days 1 and 7 revealed enrichment for genes involved with antigen presentation and mitochondrial respiratory metabolism in none/mild BPD patients, whereas in severe BPD patients, genes that changed were associated with chemokine signaling and lipid/lipoprotein metabolism. These data strongly support the role of macrophages and inflammation in the pathogenesis of BPD.

Overall, the manuscript is well written. The experiments performed are logical and the authors' conclusions are supported by their results. The study is by nature descriptive, and how these findings can be applied to the treatment of preterm infants at risk for BPD is unclear. However, the authors' findings are significant. Regarding the manuscript, I have no major concerns and only a couple minor concerns:

Major concerns: None

Minor concerns:

1. Figs 1f and 1h: Reactome analysis shows that genes in the IL-10 signaling pathway are higher in BPD at day 1 and in severe BPD at day 7. However, IL-10 levels have been reported to be decreased in tracheal aspirates and lung tissue sections from preterms who develop BPD (Oei et al, *Acta Pediatr* 2002; McColm et al, *Arch Dis Child Fetal Neonatal Ed* 2000; Garingo et al, *Pediatr Res* 2007). Although enrichment for genes in IL-10 signaling does not indicate that preterms who developed BPD had higher expression of IL-10, the authors may want to comment on this potential discrepancy.

The reviewer brings up a very good point that we have addressed in the revised manuscript. All genes differentially expressed were included in Reactome analysis. The Reactome categories shown were the top categories based on statistical significance. Importantly, Reactome categories include multiple genes within various pathways, some of which may have different functional activities. Regarding IL-10 signaling, 39 different genes are contained in the Reactome IL-10 signaling pathway. Many of these genes are pro-inflammatory mediators. The actual differentially expressed genes within our data mapping to that category include IL1A, IL1B, and CCL3, each considered a pro-inflammatory mediator. We have included this explanation in the text.

2. Previously published studies have found elevated levels of multiple proinflammatory cytokines and chemokines in tracheal aspirates from preterm infants with BPD, including IL-1B, IL-6, IL-8, IL-16, TNF α , IFN γ , CCL2, CCL7, and CCL8. Although the authors show that transcriptional profiling

confirms elevated levels of IL-1B, other cytokines and chemokines that have been reported were not found in this study. Again, the authors may want to comment on differences between other published studies and their findings.

While we did detect higher inflammatory mediator expression both on day 1 and at later time points in patients developing BPD, the reviewer is correct to note that some reported markers were not statistically elevated. The differences could be due to the mode of detection, timing of sampling, or clinical variations. We have included text in the Discussion acknowledging previous studies measuring elevated inflammatory biomarkers in BPD and the potential reasons for variability.

REVIEWERS' COMMENTS:

Reviewer #1 (Remarks to the Author):

Thanks for considering the proposed suggestions - the manuscript is largely improved.

There are still occasional typos (e.g. in the abstract, line 36) so please make sure to double check.

With regards to the term "extremely" preterm, please avoid this also in the main text unless it is an official term used in the field. In that case, please use it consistently throughout.

For the future, it would be helpful to provide a labelled version of the manuscript with highlighted changes.

Reviewer #2 (Remarks to the Author):

This manuscript investigates the role of macrophages in bronchopulmonary dysplasia (BPD). To accomplish this, transcriptome analysis using Affymetrix microarrays was used to carry out an unbiased survey of the differences in gene expression profiles in tracheal aspirate macrophages collected from intubated preterm infants. In this study, the authors show that macrophages isolated on the first day of life from patients who go on to develop BPD express higher levels of inflammatory genes such as CCL3, CCL4, CCL20, and IL23. A couple gene changes were validated using quantitative real time PCR. At day 7 of age, preterm infants that eventually developed severe BPD expressed higher levels of macrophage inflammatory mediators such as IL1A and IL1B. Pathway analysis revealed that inflammatory cytokine signaling pathways were highly represented in patients developing BPD. The authors also showed that macrophages from patients with no or mild BPD expressed a unique set of genes in the first week of life that are M2-like (high MRC1 and C1Q). Pathway analysis of genes that changed between days 1 and 7 revealed enrichment for genes involved with antigen presentation and mitochondrial respiratory metabolism in none/mild BPD patients, whereas in severe BPD patients, genes that changed were associated with chemokine signaling and lipid/lipoprotein metabolism. These data strongly support the role of macrophages and inflammation in the pathogenesis of BPD.

The revisions made by the authors have significantly improved the manuscript. Supplementary tables detailing gene lists and numbers of up-/down-regulated genes have been added. Higher resolution images have been included with the resubmission and figure legends have been revised. Additional detail regarding how microarray statistical analyses were performed have been included in the revised manuscript. The authors have also clarified which genes were differentially expressed in the pathways identified by Reactome analysis. The discussion has been expanded to acknowledge previous studies measuring elevated inflammatory biomarkers in BPD. In my opinion, all comments and suggestions supplied during the previous review have been adequately addressed. I have no major comments and only two very minor comments.

Major comments: None

Minor comments:

1. Abstract line 36 should likely say: "We propose that these changes describe for the first time..."

2. Supplementary Table 1: mean gestational age is listed as 26.3 weeks, but median gestational age is listed as 26 $\frac{6}{7}$ weeks. Recommend using either decimals or fractions to be consistent.

Response to Reviewers

Reviewer #1 (Remarks to the Author):

Thanks for considering the proposed suggestions - the manuscript is largely improved.

There are still occasional typos (e.g. in the abstract, line 36) so please make sure to double check.

We have carefully proofread the manuscript and corrected errors.

With regards to the term "extremely" preterm, please avoid this also in the main text unless it is an official term used in the field. In that case, please use it consistently throughout.

We have removed the term “extremely” when referring to preterm infants.

Reviewer #2 (Remarks to the Author):

This manuscript investigates the role of macrophages in bronchopulmonary dysplasia (BPD). To accomplish this, transcriptome analysis using Affymetrix microarrays was used to carry out an unbiased survey of the differences in gene expression profiles in tracheal aspirate macrophages collected from intubated preterm infants. In this study, the authors show that macrophages isolated on the first day of life from patients who go on to develop BPD express higher levels of inflammatory genes such as CCL3, CCL4, CCL20, and IL23. A couple gene changes were validated using quantitative real time PCR. At day 7 of age, preterm infants that eventually developed severe BPD expressed higher levels of macrophage inflammatory mediators such as IL1A and IL1B. Pathway analysis revealed that inflammatory cytokine signaling pathways were highly represented in patients developing BPD. The authors also showed that macrophages from patients with no or mild BPD expressed a unique set of genes in the first week of life that are M2-like (high MRC1 and C1Q). Pathway analysis of genes that changed between days 1 and 7 revealed enrichment for genes involved with antigen presentation and mitochondrial respiratory metabolism in none/mild BPD patients, whereas in severe BPD patients, genes that changed were associated with chemokine signaling and lipid/lipoprotein metabolism. These data strongly support the role of macrophages and inflammation in the pathogenesis of BPD.

The revisions made by the authors have significantly improved the manuscript. Supplementary tables detailing gene lists and numbers of up-/down-regulated genes have been added. Higher resolution images have been included with the resubmission and figure legends have been revised. Additional detail regarding how microarray statistical analyses were performed have been included in the revised manuscript. The authors have also clarified which genes were differentially expressed in the pathways identified by Reactome analysis. The discussion has been expanded to acknowledge previous studies measuring elevated inflammatory biomarkers in BPD. In my opinion, all comments and suggestions supplied during the previous review have been adequately addressed. I have no major comments and only two very minor comments.

Major comments: None

Minor comments:

1. Abstract line 36 should likely say: “We propose that these changes describe for the first time...”

The abstract has been edited and revised to fit *Communications Biology* style and this sentence has been removed.

2. Supplementary Table 1: mean gestational age is listed as 26.3 weeks, but median gestational age is listed as 26 6/7 weeks. Recommend using either decimals or fractions to be consistent.

Supplementary Table 1 has been revised to be consistent.